# Learning a latent manifold of odor representations from neural responses in piriform cortex

**Anqi Wu**[1]     **Stan L. Pashkovski**[2]     **Sandeep Robert Datta**[2]     **Jonathan W. Pillow**[1]

[1] Princeton Neuroscience Institute, Princeton University,
{anqiw, pillow}@princeton.edu
[2] Department of Neurobiology, Harvard Medical School,
{pashkovs, srdatta}@hms.harvard.edu

## Abstract

A major difficulty in studying the neural mechanisms underlying olfactory perception is the lack of obvious structure in the relationship between odorants and the neural activity patterns they elicit. Here we use odor-evoked responses in piriform cortex to identify a latent manifold specifying latent distance relationships between olfactory stimuli. Our approach is based on the Gaussian process latent variable model, and seeks to map odorants to points in a low-dimensional embedding space, where distances between points in the embedding space relate to the similarity of population responses they elicit. The model is specified by an explicit continuous mapping from a latent embedding space to the space of high-dimensional neural population firing rates via nonlinear tuning curves, each parametrized by a Gaussian process. Population responses are then generated by the addition of correlated, odor-dependent Gaussian noise. We fit this model to large-scale calcium fluorescence imaging measurements of population activity in layers 2 and 3 of mouse piriform cortex following the presentation of a diverse set of odorants. The model identifies a low-dimensional embedding of each odor, and a smooth tuning curve over the latent embedding space that accurately captures each neuron's response to different odorants. The model captures both signal and noise correlations across more than 500 neurons. We validate the model using a cross-validation analysis known as co-smoothing to show that the model can accurately predict the responses of a population of held-out neurons to test odorants.

## 1   Introduction

Odorants are physically described by thousands of features in a high-dimensional chemical feature space. Previous studies have focused on reducing the dimensionality of this chemical feature space [1], or on identifying dimensions of olfactory perceptual space using psychophysical measurements in humans [2, 3]. However, the dimensions of olfactory space underlying neural representations in the brain remain largely unknown. Here we take a latent variable modeling approach to the problem of identifying a low-dimensional manifold of olfactory stimuli. Our approach is unsupervised in that it makes no use of chemical features, but seeks to identify a latent embedding of odorants from measurements of odor-evoked neural population activity in mouse piriform cortex. This approach aims to provide insight into odor encoding in the brain by identifying an olfactory space that relates smoothly to changes in large-scale neural firing patterns.

Recent work in computational neuroscience has focused on the development of sophisticated model-based methods for identifying low-dimensional latent manifolds underlying neural population activity [4–12]. Here we extend such methods to the problem of neural coding in the olfactory system.

Specifically, we develop a Gaussian process based latent variable model (GPLVM) [13] for identifying latent structure underlying population activity in the olfactory cortex. The model is defined by a latent olfactory space, which serves as a low-dimensional embedding space. This latent space seeks to preserve the similarity relationships between odors on the basis of similarities in evoked neural activity patterns. The latent olfactory space is mapped to the space of high-dimensional neural activity patterns via a set of nonlinear tuning curves, one for each neuron, each governed by a Gaussian process prior. The output of these tuning curves specifies a vector of mean responses to an odorant, and we model the neural activity patterns as Gaussian with a low-rank plus diagonal covariance, modulated by an odor-dependent scaling factor. This results in a matrix normal model of the population response across odorants, defined by a diagonal odorant covariance and a low-rank plus diagonal neuron covariance matrix. The main novelty of this work from a modeling perspective consists of extending the GPLVM to incorporate structured noise for capturing correlated, odor-dependent variability in multi-trial population responses to repeated stimuli. Although we have applied it here to the piriform cortex, we feel that this model could be used to gain insights into the latent organization of neural population activity in a wide variety of other brain areas where coding is mixed or poorly understood, e.g., prefrontal cortex [14, 15], parietal cortex [16–18], or entorhinal cortex [19].

In the following, we formulate the multi-trial Gaussian process latent variable for correlated neural activity (Sec. 2) and describe an efficient variational expectation maximization (EM) inference method based on black-box variational inference (Sec. 3). We then describe a validation procedure based on co-smoothing, in which we predict the response of a subset of the neural population to a test odor using the tuning curves and the latent embeddings estimated from training data (Sec. 4). We validate our model and inference methodology using a simulated experiment, which reveals that repeated stimulus presentations are necessary to obtain accurate estimates of the structured noise covariance (Sec. 6). Finally (Sec. 7), we apply the model to multiple multi-neuron recordings of population activity from layer 2 (L2) and layer 3 (L3) mouse piriform cortex, each with more than 500 simultaneously recorded neurons. The model allows us to infer a low-dimensional embedding of 66 odorants, and smooth, low-dimensional neural tuning curves that account for the mean response of each neuron across odorants, and covariance matrices that account for both signal and noise correlations in neural activity patterns across neurons and odorants.

## 2  Multi-trial Gaussian process latent variable with structured noise

We consider simultaneously measured calcium fluorescence imaging responses from $N$ neurons in response to $D$ distinct odorants, each presented $T$ times. Let $\mathbf{Y} \in \mathbb{R}^{T \times D \times N}$ denote the tensor of neural responses, with neurons indexed by $n \in \{1, ..., N\}$, odorants indexed by $d \in \{1, ..., D\}$ and repeats indexed by $t \in \{1, ..., T\}$. Our goal is to build a generative model characterizing a low-dimensional latent structure underlying this data, and assume each odor is associated with a latent variable $\mathbf{x}_d \in \mathbb{R}^{P \times 1}$ in a $P$-dimensional latent space.

**Latent space**: Let $\mathbf{X} = [\mathbf{x}_1, ..., \mathbf{x}_D]^\top \in \mathbb{R}^{D \times P}$ denote the matrix of latent locations for the $D$ odorants in a $P$-dimensional latent embedding space. Let $\mathbf{x}_p$ denote the $p$'th column of $\mathbf{X}$, which carries the embedding location of all odorants along the $p$'th latent dimension. We place a standard normal prior to the embedding locations, $\mathbf{x}_p \sim \mathcal{N}(\mathbf{0}, \mathbf{I}_D)$ for all $p$, reflecting our lack of prior information from the chemical descriptors for each odorant.

**Nonlinear latent tuning curves**: Let $f : \mathbb{R}^{P \times 1} \to \mathbb{R}$ denote a nonlinear function mapping from the latent space of odorant embeddings $\{\mathbf{x}_d\}$ to a single neuron's firing rate. These functions differ from traditional tuning curves in that their input is the latent (unobserved) vector $\mathbf{x}_d$ of an odorant, as opposed to an observable stimulus feature (e.g., or orientation of a visual grating, or chemical features of an odorant). Let $f_n(\mathbf{x})$ denote the tuning curve for the $n$'th neuron, which we parametrize with a Gaussian Process (GP) prior:

$$f_n(\mathbf{x}) \sim \mathcal{GP}(m(\mathbf{x}), k(\mathbf{x}, \mathbf{x}')), \quad n = \{1, ..., N\} \tag{1}$$

where $m(\mathbf{x}) = \mathbf{b}_n^\top \mathbf{x}$ is a linear mean function with weights $\mathbf{b}_n$, and $k(\mathbf{x}, \mathbf{x}')$ is a covariance function that governs smoothness of the tuning curve over its $P$-dimensional input latent space. We use the Gaussian or radial basis function (RBF) covariance function: $k(\mathbf{x}, \mathbf{x}') = \rho \exp(-||\mathbf{x} - \mathbf{x}'||_2^2 / 2\sigma^2)$, where $\mathbf{x}$ and $\mathbf{x}'$ are arbitrary points in the latent space, $\rho$ is the marginal variance and $\sigma$ is the length scale controlling smoothness of the latent tuning curve.

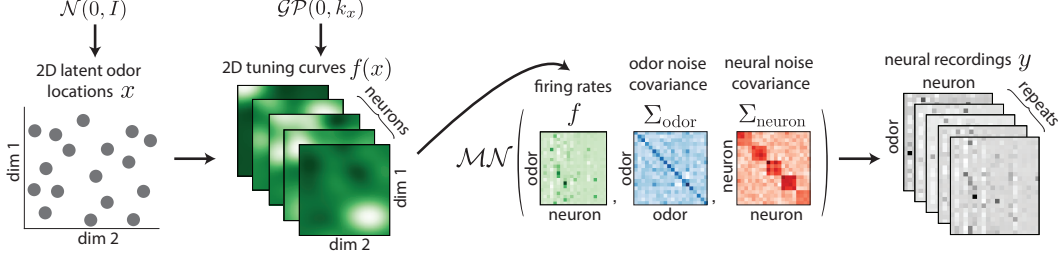

Figure 1: Schematic diagram of the multi-trial Gaussian process latent variable with structured noise.

Let $\mathbf{f}_n \in \mathbb{R}^{D \times 1}$ denote a vector of firing rates for neuron $n$ in response the $D$ odorants, with the $d$'th element equal to $f_n(\mathbf{x}_d)$. The GP prior over $f_n(\cdot)$ implies that $\mathbf{f}_n$ has a multivariate normal distribution given $\mathbf{X}$:

$$\mathbf{f}_n \mid \mathbf{X} \sim \mathcal{N}(\mathbf{m}_n, \mathbf{K}), \quad n = \{1, ..., N\} \tag{2}$$

where $\mathbf{m}_n$ is a $D \times 1$ mean vector for neuron $n$, and $\mathbf{K}$ is a $D \times D$ covariance matrix generated by evaluating the covariance function $k(\cdot, \cdot)$ at all pairs of rows in $\mathbf{X}$. We assume the mean vector to be $\mathbf{m}_n = \mathbf{X}\mathbf{b}_n$ with weights $\mathbf{b}_n \in \mathbb{R}^{P \times 1}$ giving a linearly mapping of the $P$-dimensional latent representation for the mean of the firing rate vector $\mathbf{f}_n$. If we assume a prior distribution over $\mathbf{b}_n : p(\mathbf{b}_n) = \mathcal{N}(\mathbf{0}, \beta^{-1}\mathbf{I}_P)$ for $n = \{1, ..., N\}$ with $\beta$ as the precision, we can integrate over $\mathbf{b}_n$ to get the distribution of $\mathbf{f}_n$ conditioned on $\mathbf{X}$ only:

$$\mathbf{f}_n | \mathbf{X} \sim \mathcal{N}(\mathbf{0}, \mathbf{K} + \beta^{-1}\mathbf{X}\mathbf{X}^\top), \quad n = \{1, ..., N\} \tag{3}$$

where the covariance is a mixture of a linear kernel and a nonlinear RBF kernel. The precision value $\beta$ plays a role as the trade-off parameter between two kernels. For simplicity, we will denote $\mathbf{K} + \beta^{-1}\mathbf{X}\mathbf{X}^\top$ as $\mathbf{K}$ in the following sections, and we will differentiate the RBF kernel and the mixture kernel in the experimental section. Horizontally stacking $\mathbf{f}_n$ for $N$ neurons, we get a firing rate matrix $\mathbf{F} \in \mathbb{R}^{D \times N}$ with $\mathbf{f}_n$ on the $n$'th column. Let $\widetilde{\mathbf{f}} = \text{vec}(\mathbf{F})$ be the vectorized $\mathbf{F}$, we can write the prior for $\widetilde{\mathbf{f}}$ as,

$$\widetilde{\mathbf{f}} \sim \mathcal{N}(\mathbf{0}, \mathbf{I}_N \otimes \mathbf{K}) \tag{4}$$

**Observation model**: For each repeat in the olfaction dataset, we have the neural population response to all odors, denoted as $\mathbf{Y}_t \in \mathbb{R}^{D \times N}$. Instead of taking the average over $\{\mathbf{Y}_t\}_{t=1}^T$ and modeling the averaged neural response as noise corrupted $\mathbf{F}$, we use all the repeats to estimate latent variable and noise covariance. First we collapse neuron dimension and odor dimension together to formulate a 2D matrix $\widetilde{\mathbf{Y}} \in \mathbb{R}^{T \times (DN)}$, with the row vectors $\{\widetilde{\mathbf{y}}_t \in \mathbb{R}^{(DN) \times 1}\}_{t=1}^T$. Given the vectorized firing rate $\widetilde{\mathbf{f}}$, $\{\widetilde{\mathbf{y}}_t\}_{t=1}^T$ are i.i.d samples from

$$\widetilde{\mathbf{y}}_t | \widetilde{\mathbf{f}} \sim \mathcal{N}(\widetilde{\mathbf{f}}, \boldsymbol{\Delta}), \quad t = \{1, ..., T\} \tag{5}$$

where $\boldsymbol{\Delta} \in \mathbb{R}^{(DN) \times (DN)}$ is the noise covariance matrix. When $\boldsymbol{\Delta}$ is a diagonal matrix, the model implies the observed response $y_{t,d,n} = f_{d,n} + \epsilon_{t,d,n}$ with $\epsilon_{t,d,n} \sim \mathcal{N}(0, \delta_{d,n}^2)$ for the $n$'th neuron and $d$'th odor in repeat $t$. When we assume the noise correlation exists across multiple neuron and odor pairs, $\boldsymbol{\Delta}$ is a non-diagonal matrix. In the olfaction dataset, there is a very small amount of repeats but a large neural population, which implies that $\widetilde{\mathbf{Y}}$ locates in a small-sample and high-dimension regime. Such a dataset is insufficient to provide strong data support to estimate parameters for a full rank $\boldsymbol{\Delta}$ matrix. Moreover, inverting $\boldsymbol{\Delta}$ requires $O(D^3 N^3)$ computational complexity prohibiting an efficient inference when $N$ is large. Therefore, our solution is to model the noise covariance matrix with a Kronecker structure, i.e., $\boldsymbol{\Delta} = \boldsymbol{\Sigma}_N \otimes \boldsymbol{\Sigma}_D$, where $\boldsymbol{\Sigma}_N$ is the noise covariance across neurons and $\boldsymbol{\Sigma}_D$ is the noise covariance across odors. Fig. 1 provides a schematic of the model. When applying multi-trial GPLVM to the olfactory data, each repeat of presentations of all odorants is one trial to fit the model.

**Marginal distribution over F**: Since we have normal distributions for both data likelihood (eq. 5) and prior for $\widetilde{\mathbf{f}}$ (eq. 4), we can marginalize out $\widetilde{\mathbf{f}}$ to derive the evidence for $\mathbf{X}$. There are multiple ways of deriving the integration. Here, we provide one formulation consisting of multiple multivariate

normal distributions and treating the mean and the cross-trial information as random variables:

$$p(\widetilde{\mathbf{y}}_1, ..., \widetilde{\mathbf{y}}_T | \mathbf{K}) = \mathcal{N}\left(\frac{1}{\sqrt{T}}\sum_{j=1}^{T}\widetilde{\mathbf{y}}_j | \mathbf{0}, \mathbf{\Delta} + T\mathbf{I} \otimes \mathbf{K}\right) \prod_{t=1}^{T-1} \mathcal{N}\left(\frac{1}{\sqrt{t(t+1)}}\sum_{j=1}^{t}\widetilde{\mathbf{y}}_j - \sqrt{\frac{t}{t+1}}\widetilde{\mathbf{y}}_{t+1} | \mathbf{0}, \mathbf{\Delta}\right). \quad (6)$$

More derivation details can be found in the supplement (Appendix A). The evidence distribution consists of two parts: 1) normal distributions for the cross-trial random variables with the noise covariance as its covariance, and 2) a normal distribution for the average of all repeats with a covariance formed as a sum of the noise covariance and the GP prior covariance. For single-trial data, the evidence distribution is reduced to the first normal distribution only in eq. 6, which is insufficient to be used to estimate a full noise covariance with a Kronecker structure as well as a kernel matrix. Therefore, the cross-trial statistics should be considered for structured noise estimation.

## 3 Efficient variational inference

Given the evidence in eq. 6 and the normal prior for $\mathbf{X}$, we estimate the latent variable $\mathbf{X}$ in $\mathbf{K}$ and model parameters consisting of noise covariance $\mathbf{\Delta}$ and hyperparameters in the kernel function. The joint distribution is written as,

$$p(\mathbf{Y}, \mathbf{X} | \mathbf{\Delta}, \theta) = p(\mathbf{Y} | \mathbf{X}, \mathbf{\Delta}, \theta) p(\mathbf{X}) \quad (7)$$

where $\theta = \{\rho, \sigma\}$ is the hyperparameter set, references to which will now be suppressed for simplification. This is a Gaussian process latent variable model (GPLVM) with multi-trial Gaussian observations and structured noise covariance. Due to the non-conjugacy of the data distribution and the prior over $\mathbf{X}$, we employ a variational distribution to approximate the posterior of latent variable using the Black Box Variational Inference (BBVI) [20] and optimize both latent variable and model parameters using a variational Expectation-Maximization (EM) algorithm. More details can be found in the supplement (Appendix B).

In E-step, we need to evaluate the log marginal likelihood for eq. 6 and calculate the inversion of $(DN) \times (DN)$ covariance matrices, which is the computational bottleneck of the evaluation. However, we can efficiently evaluate it with the nice property of Kronecker product. For the noise-only normal distributions, their covariance $\mathbf{\Delta} = \mathbf{\Sigma}_N \otimes \mathbf{\Sigma}_D$ is a Kronecker product of two smaller matrices. The inversion of $\mathbf{\Delta}$ is achieved by $\mathbf{\Delta}^{-1} = \mathbf{\Sigma}_N^{-1} \otimes \mathbf{\Sigma}_D^{-1}$. The log determinant is $\log|\mathbf{\Delta}| = N\log|\mathbf{\Sigma}_D| + D\log|\mathbf{\Sigma}_N|$. For the normal distribution with both latent variable and noise, its covariance matrix is a sum of two Kronecker products. In general, efficient evaluation can be carried out for such a formulation. The key idea is to transform the summation of two full matrices into one full matrix plus a diagonal matrix and then invert the summation using eigenvalue decomposition.

Let $\mathbf{\Sigma}_D = \mathbf{U}_D \mathbf{\Lambda}_D \mathbf{U}_D^\top$ and $\mathbf{\Sigma}_N = \mathbf{U}_N \mathbf{\Lambda}_N \mathbf{U}_N^\top$ be the eigen-decompositions of $\mathbf{\Sigma}_D$ and $\mathbf{\Sigma}_N$. The covariance matrix $\mathbf{C}$ can be factorized as

$$\mathbf{C} = T\mathbf{I}_N \otimes \mathbf{K} + \mathbf{\Sigma}_N \otimes \mathbf{\Sigma}_D$$
$$= \left(\mathbf{U}_N\mathbf{\Lambda}_N^{\frac{1}{2}} \otimes \mathbf{U}_D\mathbf{\Lambda}_D^{\frac{1}{2}}\right)\left((T\mathbf{\Lambda}_N^{-1}) \otimes (\mathbf{\Lambda}_D^{-\frac{1}{2}}\mathbf{U}_D^\top\mathbf{K}\mathbf{U}_D\mathbf{\Lambda}_D^{-\frac{1}{2}}) + \mathbf{I}_N \otimes \mathbf{I}_D\right)\left(\mathbf{\Lambda}_N^{\frac{1}{2}}\mathbf{U}_N^\top \otimes \mathbf{\Lambda}_D^{\frac{1}{2}}\mathbf{U}_D^\top\right). \quad (8)$$

The complexity of inverting the first and the third terms in eq. 8 is $O(D^3 + N^3)$. The bottleneck is now inverting the second term in eq. 8. We define new notations $\widetilde{\mathbf{K}} = \mathbf{\Lambda}_D^{-\frac{1}{2}}\mathbf{U}_D^\top\mathbf{K}\mathbf{U}_D\mathbf{\Lambda}_D^{-\frac{1}{2}}$ and $\widetilde{\mathbf{C}} = T\mathbf{\Lambda}_N^{-1} \otimes \widetilde{\mathbf{K}} + \mathbf{I}_N \otimes \mathbf{I}_D$.

The problem is thus reduced to inverting the matrix $\widetilde{\mathbf{C}}$. The second step is to exploit the compatibility of a Kronecker product plus a constant diagonal term with eigenvalue decomposition. Let $T\mathbf{\Lambda}_N^{-1} = \mathbf{U}_T\mathbf{\Lambda}_T\mathbf{U}_T^\top$ and $\widetilde{\mathbf{K}} = \mathbf{U}_K\mathbf{\Lambda}_K\mathbf{U}_K^\top$ be the eigen-decompositions of $T\mathbf{\Lambda}_N^{-1}$ and $\widetilde{\mathbf{K}}$. Thus,

$$\widetilde{\mathbf{C}} = T\mathbf{\Lambda}_N^{-1} \otimes \widetilde{\mathbf{K}} + \mathbf{I}_N \otimes \mathbf{I}_D = (\mathbf{U}_T \otimes \mathbf{U}_K)(\mathbf{\Lambda}_T \otimes \mathbf{\Lambda}_K + \mathbf{I}_N \otimes \mathbf{I}_D)(\mathbf{U}_T^\top \otimes \mathbf{U}_K^\top), \quad (9)$$

Finally, combining eq. 8 and eq. 9 to get

$$\mathbf{C} = \left(\mathbf{U}_N\mathbf{\Lambda}_N^{\frac{1}{2}} \otimes \mathbf{U}_D\mathbf{\Lambda}_D^{\frac{1}{2}}\right)(\mathbf{U}_T \otimes \mathbf{U}_K)(\mathbf{\Lambda}_T \otimes \mathbf{\Lambda}_K + \mathbf{I}_N \otimes \mathbf{I}_D)(\mathbf{U}_T^\top \otimes \mathbf{U}_K^\top)\left(\mathbf{\Lambda}_N^{\frac{1}{2}}\mathbf{U}_N^\top \otimes \mathbf{\Lambda}_D^{\frac{1}{2}}\mathbf{U}_D^\top\right). \quad (10)$$

Inverting $\mathbf{C}$ now has only $O(D^3 + N^3)$ computational complexity instead of $O(D^3 N^3)$. More detailed derivations can be found in the supplement (Appendix C). With this efficient evaluation of the log conditional likelihood, we can run BBVI fast for E-step to learn the optimal approximate posterior $q(\mathbf{X}|\lambda^\dagger) \approx p(\mathbf{X}|\mathbf{Y}, \boldsymbol{\Delta}, \theta)$ given a fixed set of $\boldsymbol{\Delta}$ and $\theta$ with $\lambda^\dagger$ as the optimal approximation parameters.

In M-step, model parameters are optimized using the ELBO given the optimal variational distribution learned from E-step:

$$\boldsymbol{\Delta}^\dagger, \theta^\dagger = \text{argmax}_{\boldsymbol{\Delta}, \theta} \, \mathbb{E}_{q(\mathbf{X}|\lambda^\dagger)} \left[ \log p(\mathbf{Y}|\mathbf{X}, \boldsymbol{\Delta}, \theta) \right] \tag{11}$$

where the expectation can also be approximated by Monte Carlo integration.

After the optimization, we can derive the posterior distribution for firing rates $\mathbf{F}$ given the neural response $\mathbf{Y}$ and optimal $\mathbf{X}$, $\boldsymbol{\Delta}$ and $\theta$ as

$$p(\mathbf{F}|\mathbf{Y}, \mathbf{X}, \boldsymbol{\Delta}, \theta) = \mathcal{N}\left( \widetilde{\mathbf{f}} | (\mathbf{I}_N \otimes \mathbf{K})(\boldsymbol{\Delta} + T\mathbf{I}_N \otimes \mathbf{K})^{-1} \sum_{t=1}^{T} \widetilde{\mathbf{y}}_t, (\mathbf{I}_N \otimes \mathbf{K})(\boldsymbol{\Delta} + T\mathbf{I}_N \otimes \mathbf{K})^{-1}\boldsymbol{\Delta} \right). \tag{12}$$

Similar to the evaluation in E-step, the posterior mean of firing rates can be efficiently calculated using the same Kronecker trick in eq. 10.

# 4 Prediction with co-smoothing

We propose a model to learn latent representations for odors and tuning curves for neurons as well as structured noise covariance with multi-trial neural responses. Next, we employ a *co-smoothing* idea to evaluate its performance. The question to ask is when presenting an unseen odor to neural populations, can we use partially observed neurons' responses to learn the odor's latent representation, then predict the neural responses of the unobserved neurons given their tuning curves and the latent representation?

**Firing rate prediction**: We first use the training odors to estimate the firing rates and the latent representations of these training odors as shown in sec. 3. For a new odor, we collect some repeats of neural responses from partially observed neural ensembles $\mathbf{Y}_o^* \in \mathbb{R}^{T \times 1 \times N_o}$ where $T$ is the number of repeats, $N_o$ is the number of observed neurons and $*$ indicates the test odor. We use $\mathbf{Y}_o^*$ as well as the optimal firing rates $\mathbf{F}$ and latent variables $\mathbf{X}$ to estimate the latent representation $\mathbf{x}^*$ for the test odor. We use the same variational EM algorithm to learn $q(\mathbf{x}^*) \approx p(\mathbf{x}^*|\mathbf{Y}_o^*, \mathbf{Y}, \mathbf{X}, \boldsymbol{\Delta}, \theta)$ by fixing the latent variables and noise covariance from the training data as well as the hyperparameters while changing the latent variable and noise variance related to the test odor. Finally, the predictive firing rate for the test odor from the partially unobserved neural ensembles, denoted as $\mathbf{F}_u^* \in \mathbb{R}^{N_u \times 1}$ with $N_u$ as the number of unobserved neurons, is calculated as

$$\mathbf{F}_u^* = (\boldsymbol{\Gamma}_{N_u, N} \otimes \mathbf{K}^*)(\boldsymbol{\Delta} + T\mathbf{I}_N \otimes \mathbf{K})^{-1} \sum_{t=1}^{T} \widetilde{\mathbf{y}}_t, \tag{13}$$

where $\mathbf{K}^* \in \mathbb{R}^{1 \times D}$ is the kernel matrix evaluated between the test odor's latent representation $\mathbf{x}^*$ and the training odors' latent representations $\mathbf{X}$, and $\boldsymbol{\Gamma}_{N_u, N} \in \mathbb{R}^{N_u \times N}$ is a zero-one matrix indicating the indices of the unobserved neurons in the entire neural ensemble. We can also calculate the firing rates for the observed neurons $\mathbf{F}_o^* \in \mathbb{R}^{N_o \times 1}$ using a similar expression as eq. 13. For experimental evaluation purpose, we can compare the predictive firing rate $\mathbf{F}_u^*$ with the averaged true response $\sum_{t=1}^{T} \mathbf{Y}_{u,t}^*$. We will show the firing rate prediction in the olfaction data experiment.

**Single-trial neural activity prediction**: When the number of repeats is large enough to render a mean response resembling the underlying firing rate, single-trial and trial-average models can both provide good estimations for latent variables and firing rates for test odors by using the co-smoothing approach. The advantage of our multi-trial model will be suppressed when only evaluating the predictive performance for firing rates when there are many repeats. Thereby, we can take another step forward to predict single-trial neural activities given the estimated firing rates where the estimated noise covariance encodes trial-by-trial deviations from the noise-free firing rate.

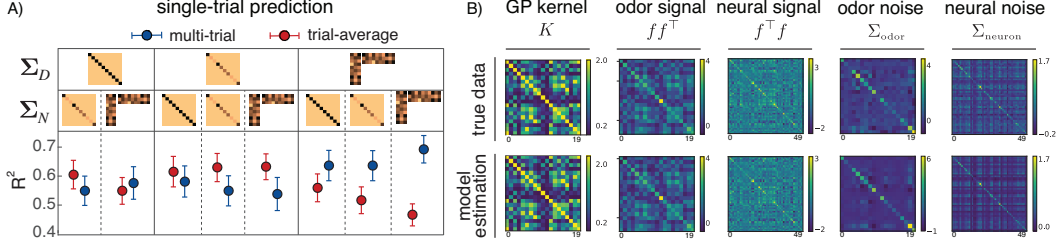

Figure 2: A) $R^2$ values for single-trial prediction for 8 different noise covariance structures comparing between trial-average and multi-trial models. The top two rows indicate the combinations of neuron noise covariance and odor noise covariance parametrization. The y-axis indicates the $R^2$ values. B) Data covariance/correlation (top) and model-recovered covariance/correlation (bottom) for signal (columns 2-3) and noise (columns 4-5). The true kernel matrix in the GP prior is presented at the top in the 1st column and the estimated kernel matrix is presented at the bottom in the 1st column.

Let $\boldsymbol{\Sigma}_D$ and $\boldsymbol{\Sigma}_N$ denote the noise covariance matrices for all odors and all neurons. We can partition them into the following forms:

$$\boldsymbol{\Sigma}_D = \begin{bmatrix} \boldsymbol{\Sigma}_D^{11} & \boldsymbol{\Sigma}_D^{12} \\ \boldsymbol{\Sigma}_D^{12\top} & \boldsymbol{\Sigma}_D^{22} \end{bmatrix}, \quad \boldsymbol{\Sigma}_N = \begin{bmatrix} \boldsymbol{\Sigma}_N^{11} & \boldsymbol{\Sigma}_N^{12} \\ \boldsymbol{\Sigma}_N^{12\top} & \boldsymbol{\Sigma}_N^{22} \end{bmatrix}. \tag{14}$$

$\boldsymbol{\Sigma}_D$ is partitioned according to the training odors and the test odor. $\boldsymbol{\Sigma}_D^{11}$ is the noise covariance for the training odors estimated during the training stage; $\boldsymbol{\Sigma}_D^{12}$ is the cross noise covariance between the training odors and the test odor estimated during the co-smoothing stage; and $\boldsymbol{\Sigma}_D^{22}$ is the test odor noise covariance estimated during the co-smoothing stage. $\boldsymbol{\Sigma}_N$ is partitioned according to the observed neurons and the unobserved neurons. $\boldsymbol{\Sigma}_N^{11}$ is the noise covariance for the observed odors; $\boldsymbol{\Sigma}_N^{12}$ is the cross noise covariance between the observed neurons and the unobserved neurons; and $\boldsymbol{\Sigma}_N^{22}$ is the unobserved neuron noise covariance. The entire $\boldsymbol{\Sigma}_N$ matrix is learned during the training procedure and is partially used to do co-smoothing. We also denote the single-trial neural response for training as $\mathbf{Y}_t$, the single-trial neural response added for co-smoothing as $\mathbf{Y}_{o,t}^*$ and the single-trial neural response from the unobserved neurons for the test odor as $\mathbf{Y}_{u,t}^*$. Then we can write down the mean of the posterior distribution for $\mathbf{Y}_{u,t}^*$, i.e., $p(\mathbf{Y}_{u,t}^*|\mathbf{Y}_t, \mathbf{Y}_{o,t}^*, \mathbf{F}, \mathbf{F}_o^*, \mathbf{F}_u^*, \boldsymbol{\Sigma}_D, \boldsymbol{\Sigma}_N)$, as

$$\text{vec}\hat{\mathbf{Y}}_{u,t}^* = \text{vec}\hat{\mathbf{F}}_u^* + \begin{bmatrix} \boldsymbol{\Sigma}_D^{12}\otimes\begin{bmatrix} \boldsymbol{\Sigma}_N^{12} \\ \boldsymbol{\Sigma}_N^{22} \end{bmatrix} \\ \boldsymbol{\Sigma}_D^{22}\otimes\boldsymbol{\Sigma}_N^{12} \end{bmatrix}^\top \begin{bmatrix} \boldsymbol{\Sigma}_D^{11}\otimes\boldsymbol{\Sigma}_N & \boldsymbol{\Sigma}_D^{12}\otimes\begin{bmatrix} \boldsymbol{\Sigma}_N^{11} \\ \boldsymbol{\Sigma}_N^{12\top} \end{bmatrix} \\ \boldsymbol{\Sigma}_D^{12\top}\otimes[\boldsymbol{\Sigma}_N^{11} \ \boldsymbol{\Sigma}_N^{12}] & \boldsymbol{\Sigma}_D^{22}\otimes\boldsymbol{\Sigma}_N^{11} \end{bmatrix}^{-1} \begin{bmatrix} \text{vec}\mathbf{Y}_t - \text{vec}\mathbf{F} \\ \text{vec}\mathbf{Y}_{o,t}^* - \text{vec}\mathbf{F}_o^* \end{bmatrix} \tag{15}$$

We will show the predictive performance comparing $\hat{\mathbf{Y}}_{u,t}^*$ and $\mathbf{Y}_{u,t}^*$ using single repeats in the simulated experiment.

## 5 Simulated data

First, we consider a simulated dataset to illustrate the effect of our multi-trial GPLVM model with structured noise covariance on single-trial predictive performance. We create a simulated example with $T = 10$ repeats, $N = 50$ neurons and $D = 20$ odors according to the generative model described in sec. 2. We generate 2-dimensional latent variables from a normal prior and construct a covariance matrix from the latent using an RBF kernel function, and then i.i.d sample tuning curves for 50 neurons from a Gaussian process prior with a zero mean and the covariance matrix. Then we generate two structured noise covariance matrices with $rank = 2$ for neurons and odors respectively. Finally, we generate 10 samples from eq. 5 using the sampled tuning curves and the structured noise covariances.

We compare multiple combinations of structures for neuron noise covariance $\boldsymbol{\Sigma}_N$ and odor noise covariance $\boldsymbol{\Sigma}_D$. Each one can take one of three forms: an identity matrix, a diagonal matrix with heterogeneous noise variances on the diagonal and a low-rank full matrix plus a heterogeneous diagonal (indicated in Fig. 2A)). Moreover, we compare between trial-averaged neural response and multi-trial neural response in order to show that it requires more statistics to learn structured noise variance. The trial-average results in Fig. 2A) are achieved by fitting the mean response only to

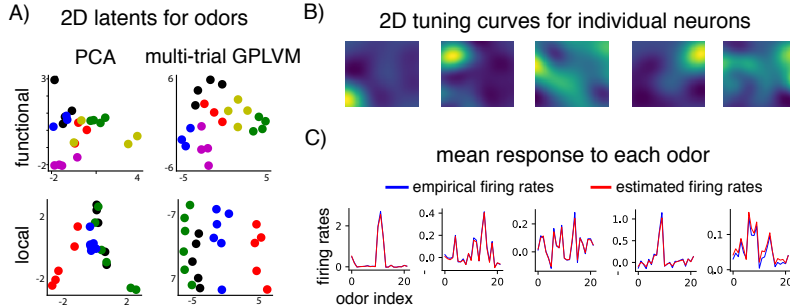

Figure 3: A) 2D latent representations of 22 odors in functional and local odor sets analyzed by PCA and multi-trial GPLVM. Odors from different functional groups are color-coded. B) Inferred two-dimensional latent tuning curves for five example neurons. C) Mean response to each of the 22 individual odors for these same example neurons. Traces show observed mean spike count for each odor (blue) and inferred latent tuning curve value (red).

multi-trial GPLVM to learn structured noise. Our quantitative comparison covers the noise models for GP from [21] and [22]. The $R^2$ values of the single-trial prediction performance is shown in Fig. 2A). The red and blue error bars represent trial-average and multi-trial respectively. When fitting a full noise covariance matrix for odors, a trial-average model is poor. When fitting the 8th column with full matrices for both neurons and odors, it prefers the multi-trial model and achieves the best predictive performance with structured noise covariance matrices. We also show that the best model (the 8th column) effectively captures noise structures and signal structures for both neuron and odor from the data (Fig. 2B)). The kernel matrix for the prior is also well recovered in Fig. 2B).

## 6 Olfaction data

Two-photon calcium imaging of piriform cortex was performed in awake mice previously infected with the GCaMP6s activity reporter. Imaging volumes through piriform layers 2 and 3 were acquired at 7 volumes/sec using a custom microscope equipped with a resonant galvo and high-speed piezo actuator. Detection of active neurons, segmentation, and extraction of fluorescence signal was performed using Suite2p software. Extracted fluorescence traces were corrected for neuropil contamination. For each cell, responses to odor presentations constituted a single delta F/F0 value where F is the average fluorescence signal over 2 seconds immediately following odor onset and F0 is fluorescence signal preceding odor onset. Monomolecular odors were diluted in di-propylene glycol (DPG) according to individual vapor pressures obtained from www.thegoodscentscompany.com, to give a nominal concentration of 500 ppm. This vapor-phase concentration was further diluted 1:5 by the carrier airflow to yield 100 ppm at the exit port. Odor presentations lasted for two seconds and were interleaved by 30 seconds of blank (DPG) delivery. The order of presentation of odors was pseudo-randomized for each experiment, such that on any given repeat, odors were presented once in no predictable order. Three different odor sets, each consisting of 22 odorants, were presented to multiple awake mice with 10 repeats for each odor. For each odor set, we have calcium imaging neural responses collected from about 200 neurons in both layer 2 (L2) and layer 3 (L3) in the piriform cortex of 3 mice leading to a dataset with about 500 L2 neurons and 500 L3 neurons for each odor set. Therefore, we deal with three datasets, each with $T = 10$ repeats, $D = 22$ odorants, $N \approx 500$ L2 neurons and $N \approx 500$ L3 neurons.

We standardize each repeat response across neurons and apply principle component analysis (PCA) and our model with a 2-dimensional latent embedding to these datasets. For PCA, we find the first two principal components of the $D \times (NT)$ response matrix. For our model, the kernel in eq. 3 is an RBF function without a linear component. We set the noise covariances for odors and neurons to be a heterogeneous diagonal matrix and a full matrix with a low-rank structure as described in Fig. 2A). We fit the model to three different odor sets {"functional", "local", "global"} using both L2 neurons and L3 neurons sharing the same 2D latent variables. Fig. 3A) shows the 2-dimensional latent variables for 22 odors in the functional and local odor sets. More latent representations discovered by t-SNE [23] and multidimensional scaling (MDS) [24] can be found in the supplementary (Appendix D).

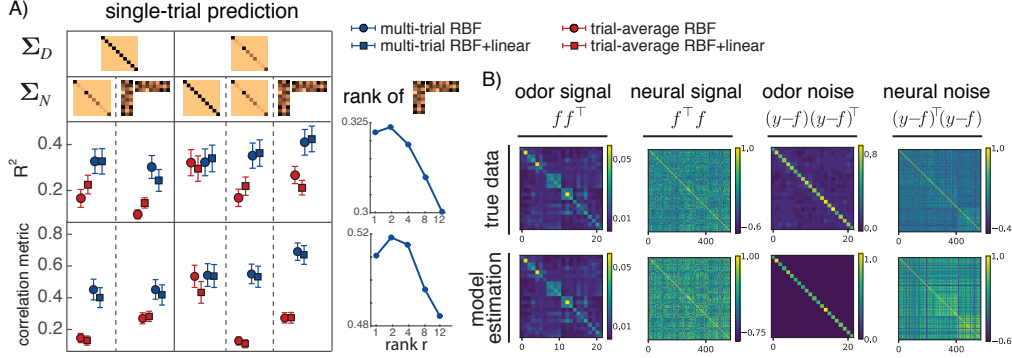

Figure 4: A) $R^2$ and correlation metric criteria for predictive performance for 5 different noise covariance structures comparing between trial-average and multi-trial models as well as an RBF kernel vs a mixture of kernels. The top two rows indicate the combinations of neuron noise structure and odor noise structure. The influence of the rank of noise covariance is also presented for two criteria. B) Data covariance/correlation (top) and model-recovered covariance/correlation (bottom) for signal (the first two columns) and noise (the last two columns).

The functional odor set contains distinct odors sharing one of six chemical functional groups. Odors sharing the same functional group should be more closely related in chemical space than odors harboring different functional groups. The local odor set contains straight chain aliphatic odorants that harbor 1 of 4 carbonyl functional groups and range 3-8 carbons in length. PCA cannot discover the functional class nor identify the linearized embeddings effectively for both sets. Our model (multi-trial GPLVM) can identify 2-dimensional clusters with clear linear boundaries for the functional set and linearized curves of groups of odors for the local set, without knowing any information regarding the chemical features (Fig. 3A)). Odors from the same functional group have the same color. We learn the 2D latent variables by imposing L2 and L3 sharing the same latent space, but the tuning curves are estimated separately with different length scales for the GP priors. We observe that L3 neurons have a bigger length scale value than L2 neurons. This implies wider tuning curves for L3 which leads to better performance for L3 at discriminating different functional groups and identifying the latent odor embeddings. Fig. 3B) shows some example 2D tuning curves from L3 in both odor sets. Fig. 3C) presents averaged firing rates for individual neurons. The blue curves are the mean responses across repeats which can be considered as empirical tuning curves (signal). The red curves are estimated tuning curves. This comparison suggests that our model can identify the signal and fit the data pretty well. Moreover, 1D empirical curves are plotted along the indices of the odors which are not that smooth nor interpretable. We can see that the model can effectively capture a set of smooth 2D neural turning curves for individual neurons which explicitly map the 2D latent representations of odors to high-dimensional neural activities.

The 2D illustration indicates the strength of our proposed model in discovering nonlinear latent embedding for neural ensembles. We can find more interpretable 2D tuning curves than just taking the average across multiple repeats for single neurons. Thereby, such a 2D space can be interpreted as an underlying embedding of neural populations. Next, we will employ the co-smoothing idea described in sec. 4 to evaluate the predictive power of our model with different noise structures. The better the predictive performance is, the better the data is fit and explained by the noise structure. For evaluating purpose, we leave one odor out for each odor set, train on 21 odors using L3 neurons and compute the predicted neural activities, an $N_u$ by 1 vector, for the test odor within the odor set. In total, we carry out a training and predicting procedure for 66 times (leaving one odor out at each time) and take the average. Given the predicted neural activity vector, we use two evaluating criteria: r-squared value ($R^2$) and correlation metric. $R^2$ reveals how close the true neural activities are to the predicted ones. It emphasizes single-neuron performance. However neurons in the piriform cortex are known for encoding correlation information of odors at the population level rather than individual neurons. The correlation/similarity between odors represented in neural space is more informative. We propose another correlation-based metric. We compare the correlation between the predicted neural activity of the test odor and the training odors to obtain a 21 by 1 vector and compare this vector with the true 21 by 1 vector constructed from the true neural activities using another r-squared comparison. This is saying whether the similarity between the test odor and the

training odors estimated by the model resembles the true correlation in neural space. The correlation metric should have higher r-squared values than $R^2$ employed on the predictive neural activity vector since noisy neurons are smoothed out in the correlation metric.

Fig. 4A) presents both $R^2$ and correlation metric (y-axis) on 5 different noise models. For both metrics, the higher the y value is, the better the performance is. The structures of the models are indicated in the top two rows. When fitting the olfaction data, we don't assume a low-rank matrix for odor noise covariance. Since the presentation of odors were randomized, odors across repeats don't imply each other. The red and blue error bars represent trial-average and multi-trial respectively. It's clear that trial-average has much poor performance, especially for non-identity $\mathbf{\Sigma}_N$ matrices. When $\mathbf{\Sigma}_N$ is an identity matrix (the 3th column), the trial-average values almost catch up with the multi-trial performance. The circle represents a single RBF kernel, and the square is a mixture of RBF and linear kernels with precision $\beta$ estimated as an element in the hyperparameter set. Among all the models, the 5th model outperforms the others with a full-matrix $\mathbf{\Sigma}_N$ and a non-identity $\mathbf{\Sigma}_D$. This essentially suggests that there exists correlated noise variability among neurons which cannot be ignored and contribute to information encoding in the piriform cortex. Odorants are more independent in neural space but require odor-specific noise variances. This result validates our prior knowledge about the olfactory neurons. Fig. 4B) shows that the best model (the 5th column) effectively captures noise structures and signal structures for both neuron and odor from the data.

There are two dimensionality parameters we need to tune in the model. One is the dimensionality of the latent space, and the other is the rank of the low-rank component in the structured noise matrix. We automate the selection of the number of latent dimensions via an automatic relevance determination (ARD) kernel [25] version of RBF over the latent variables, i.e. $\mathbf{K}$ in eq. 2 achieved by $k(\mathbf{x}, \mathbf{x}') = \rho \exp(- \sum_i^P \frac{(\mathbf{x}_i - \mathbf{x}_i')^2}{2\sigma_i^2})$. Each latent dimension has its own length scale $\sigma_i^2$, and they are independent of each other. By fitting the length scale $\sigma_i^2$, the model automatically learns a sparse latent space with most $\sigma_i^2$s approaching to infinity and a few small $\sigma_i^2$s. As a result of ARD, irrelevant latent dimensions are effectively turned off by selecting large length scales for them. We initially set the dimensionality to be 100, and the model returns 10-15 effective dimensions for all the data. For the rank $r$ of the low-rank structure, we run experiments with $r = \{1, 2, 4, 8, 12\}$. Fig. 4A) shows that $r = 2$ has the best predictive performance using both $R^2$ and correlation metric suggesting the noise correlation is pretty strong with a low-dimensional subspace.

## 7 Conclusion

We have proposed a multi-trial Gaussian process latent variable model with structured noise, and used it to infer a latent odor manifold underlying olfactory responses in the piriform cortex. The resulting model maps odorants to points in a low-dimensional embedding space, where the distance between points in this embedding space relates to the similarity of population responses they elicit. The model is specified by an explicit continuous mapping from a latent embedding space to the space of high-dimensional neural population activity patterns via a set of nonlinear neural tuning curves, each parametrized by a Gaussian process, followed by a low-rank model of correlated, odor-dependent Gaussian noise. We used multiple repeats for analysis instead of trial-average responses for the sake of structured noise covariance estimation. We applied this model to calcium fluorescence imaging measurements of population activity in layers 2 and 3 of mouse piriform cortex following presentation of a diverse set of odorants. We showed that we can learn a low-dimensional embedding of odorants and a smooth tuning curve over the latent embedding space that accurately captures neural responses to different odorants. The model captured both signal and noise correlations across more than 500 neurons. Finally, we performed a co-smoothing analysis to show that the model can accurately predict responses of a population of held-out neurons to test odorants.

In the future, we will further investigate the biological interpretability of the 10-15 effective latent dimensions for olfactory perceptual space and the rank-2 structured neural noise covariance. Moreover, we will explore the relationship between chemical features of these odorants and their learned latent embeddings in order to understand which chemical features are most important for determining an odorant's location within the neural manifold for olfactory representations.

## Acknowledgements

This work was supported by grants from the Simons Foundation (SCGB AWD1004351 and AWD543027), the NIH (R01EY017366, R01NS104899) and a U19 NIH-NINDS BRAIN Initiative Award (NS104648-01).

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
