[Supplementary Material]

## Appendix A: derivation for the marginal distribution in eq. 6

First, we show a general derivation for the marginal distribution over $\mathbf{F}$. First, we concatenate $\{\widetilde{\mathbf{y}}_t\}_{t=1}^{T}$ vertically to get a big vector $\hat{\mathbf{y}} \in \mathbb{R}^{(TDN)\times 1}$ resulting in the following distribution

$$\hat{\mathbf{y}}|\widetilde{\mathbf{f}} \sim \mathcal{N}(\mathbf{h} \otimes \widetilde{\mathbf{f}}, \mathbf{I}_T \otimes \mathbf{\Delta}) \tag{16}$$

where $\mathbf{h}$ is a $T \times 1$ vector of ones. We can replace $\mathbf{h}$ with a $(TDN) \times (DN)$ matrix $\mathbf{H}$ such that $\mathbf{H}\widetilde{\mathbf{f}} = \mathbf{h} \otimes \widetilde{\mathbf{f}}$. The marginal distribution of $\hat{\mathbf{y}}$ can be written as

$$
\begin{aligned}
p(\hat{\mathbf{y}}|\mathbf{K}) &= \int \mathcal{N}(\hat{\mathbf{y}}|\mathbf{H}\widetilde{\mathbf{f}}, \mathbf{I}_T \otimes \mathbf{\Delta})\mathcal{N}(\widetilde{\mathbf{f}}|\mathbf{0}, \mathbf{I}_N \otimes \mathbf{K})d\widetilde{\mathbf{f}} \\
&= \frac{1}{Z} \int \exp\left(-\frac{1}{2}(\hat{\mathbf{y}} - \mathbf{H}\widetilde{\mathbf{f}})^\top (\mathbf{I}_T \otimes \mathbf{\Delta})^{-1}(\hat{\mathbf{y}} - \mathbf{H}\widetilde{\mathbf{f}}) - \frac{1}{2}\widetilde{\mathbf{f}}^\top(\mathbf{I}_N \otimes \mathbf{K})^{-1}\widetilde{\mathbf{f}}\right)d\widetilde{\mathbf{f}} \\
&= \frac{1}{Z'}\exp\left(-\frac{1}{2}\hat{\mathbf{y}}^\top\left(\mathbf{I}_T \otimes \mathbf{\Delta} + \mathbf{H}(\mathbf{I}_N \otimes \mathbf{K})\mathbf{H}^\top\right)^{-1}\hat{\mathbf{y}}\right) \\
&= \mathcal{N}(\hat{\mathbf{y}}|\mathbf{0}, \mathbf{I}_T \otimes \mathbf{\Delta} + \mathbf{H}(\mathbf{I}_N \otimes \mathbf{K})\mathbf{H}^\top)
\end{aligned}
\tag{17}
$$

The new covariance $\mathbf{I}_T \otimes \mathbf{\Delta} + \mathbf{H}(\mathbf{I}_N \otimes \mathbf{K})\mathbf{H}^\top$ is a $(TDN) \times (TDN)$ matrix which is computationally not invertible in practice. However the heavy inversion can be resolved by applying matrix inversion lemma and the property of Kronecker product when calculating the log-likelihood.

Here, we provide another way of marginalizing out $\mathbf{F}$ which consists of multiple multivariate normal distributions with smaller scale covariance matrices. Instead of vectorizing $\widetilde{\mathbf{Y}}$ matrix into $\hat{\mathbf{y}}$ and dealing with one multivariate normal with a big covariance matrix, we work on the integration with the Gaussian distribution for data in eq. 5. The marginal distribution can be written as

$$p(\widetilde{\mathbf{y}}_1, ..., \widetilde{\mathbf{y}}_T|\mathbf{K}) = \int \mathcal{N}(\widetilde{\mathbf{f}}|\mathbf{0}, \mathbf{I}_N \otimes \mathbf{K})\prod_{t=1}^{T}\mathcal{N}(\widetilde{\mathbf{y}}_t|\widetilde{\mathbf{f}}, \mathbf{\Delta})d\widetilde{\mathbf{f}} \tag{18}$$

Given a set of data observations $\{\widetilde{\mathbf{y}}_t\}_{t=1}^{T}$, we can write the probability density function of $\mathcal{N}(\widetilde{\mathbf{y}}_t|\widetilde{\mathbf{f}}, \mathbf{\Delta})$ as $\mathcal{N}(\widetilde{\mathbf{f}}|\widetilde{\mathbf{y}}_t, \mathbf{\Delta})$ which is just an exponential function of a negative quadratic function. According to the property of the product of Gaussian densities, let $\mathcal{N}_x(m, \Sigma)$ denote a density of $x$, then

$$
\begin{aligned}
\mathcal{N}_x(m_1, \mathbf{\Sigma}_1)\mathcal{N}_x(m_2, \mathbf{\Sigma}_2) &= c_c\mathcal{N}_x(m_c, \mathbf{\Sigma}_c), & c_c &= \mathcal{N}_{m_1}(m_2, \mathbf{\Sigma}_1 + \mathbf{\Sigma}_2), \\
m_c &= (\mathbf{\Sigma}_1^{-1} + \mathbf{\Sigma}_2^{-1})^{-1}(\mathbf{\Sigma}_1^{-1}m_1 + \mathbf{\Sigma}_2^{-1}m_2), & \mathbf{\Sigma}_c &= (\mathbf{\Sigma}_1^{-1} + \mathbf{\Sigma}_2^{-1})^{-1}.
\end{aligned}
\tag{19}
$$

We can apply the property to the integration in eq. 18 in a chain style from $\mathcal{N}(\widetilde{\mathbf{f}}|\widetilde{\mathbf{y}}_1, \mathbf{\Delta})$ all the way to $\mathcal{N}(\widetilde{\mathbf{f}}|\mathbf{0}, \mathbf{I}_N \otimes \mathbf{K})$:

$(1).\qquad P_1 = \mathcal{N}_{\widetilde{\mathbf{f}}}(\widetilde{\mathbf{y}}_1, \boldsymbol{\Delta})\mathcal{N}_{\widehat{\mathbf{f}}}(\widetilde{\mathbf{y}}_2, \boldsymbol{\Delta}) = c_1 \mathcal{N}_{\widehat{\mathbf{f}}}\left(\frac{1}{2}(\widetilde{\mathbf{y}}_1 + \widetilde{\mathbf{y}}_2), \frac{1}{2}\boldsymbol{\Delta}\right),$

$\qquad\qquad c_1 = \mathcal{N}_{\widetilde{\mathbf{y}}_1}(\widetilde{\mathbf{y}}_2, 2\boldsymbol{\Delta}),$

$(2).\qquad P_2 = c_1 \mathcal{N}_{\widehat{\mathbf{f}}}\left(\frac{1}{2}(\widetilde{\mathbf{y}}_1 + \widetilde{\mathbf{y}}_2), \frac{1}{2}\boldsymbol{\Delta}\right)\mathcal{N}_{\widehat{\mathbf{f}}}(\widetilde{\mathbf{y}}_3, \boldsymbol{\Delta}) = c_1 c_2 \mathcal{N}_{\widehat{\mathbf{f}}}\left(\frac{1}{3}(\widetilde{\mathbf{y}}_1 + \widetilde{\mathbf{y}}_2 + \widetilde{\mathbf{y}}_3), \frac{1}{3}\boldsymbol{\Delta}\right),$

$\qquad\qquad c_2 = \mathcal{N}_{\frac{1}{2}(\widetilde{\mathbf{y}}_1 + \widetilde{\mathbf{y}}_2)}(\widetilde{\mathbf{y}}_3, \frac{3}{2}\boldsymbol{\Delta}),$

$(3).\qquad P_3 = c_1 c_2 \mathcal{N}_{\widehat{\mathbf{f}}}\left(\frac{1}{3}(\widetilde{\mathbf{y}}_1 + \widetilde{\mathbf{y}}_2 + \widetilde{\mathbf{y}}_3), \frac{1}{3}\boldsymbol{\Delta}\right)\mathcal{N}_{\widehat{\mathbf{f}}}(\widetilde{\mathbf{y}}_4, \boldsymbol{\Delta}) = c_1 c_2 c_3 \mathcal{N}_{\widehat{\mathbf{f}}}\left(\frac{1}{4}(\widetilde{\mathbf{y}}_1 + \widetilde{\mathbf{y}}_2 + \widetilde{\mathbf{y}}_3 + \widetilde{\mathbf{y}}_4), \frac{1}{4}\boldsymbol{\Delta}\right),$

$\qquad\qquad c_3 = \mathcal{N}_{\frac{1}{3}(\widetilde{\mathbf{y}}_1 + \widetilde{\mathbf{y}}_2 + \widetilde{\mathbf{y}}_3)}(\widetilde{\mathbf{y}}_4, \frac{4}{3}\boldsymbol{\Delta}),$

$\qquad\qquad \vdots$

$(T-1).\qquad P_{T-1} = \prod_{t=1}^{T-2} c_t \mathcal{N}_{\widehat{\mathbf{f}}}\left(\frac{1}{T-1}\sum_{t=1}^{T-1}\widetilde{\mathbf{y}}_t, \frac{1}{T-1}\boldsymbol{\Delta}\right)\mathcal{N}_{\widehat{\mathbf{f}}}(\widetilde{\mathbf{y}}_T, \boldsymbol{\Delta})\prod_{t=1}^{T-1} c_t \mathcal{N}_{\widehat{\mathbf{f}}}\left(\frac{1}{T}\sum_{t=1}^{T}\widetilde{\mathbf{y}}_t, \frac{1}{T}\boldsymbol{\Delta}\right),$

$\qquad\qquad c_{T-1} = \mathcal{N}_{\frac{1}{T-1}\sum_{t=1}^{T-1}\widetilde{\mathbf{y}}_t}(\widetilde{\mathbf{y}}_T, \frac{T}{T-1}\boldsymbol{\Delta}),$

$(T).\qquad P_T = \prod_{t=1}^{T-1} c_t \mathcal{N}_{\widehat{\mathbf{f}}}\left(\frac{1}{T}\sum_{t=1}^{T}\widetilde{\mathbf{y}}_t, \frac{1}{T}\boldsymbol{\Delta}\right)\mathcal{N}_{\widehat{\mathbf{f}}}(\mathbf{0}, \mathbf{I}_N \otimes \mathbf{K}) = \prod_{t=1}^{T} c_t \mathcal{N}_{\widehat{\mathbf{f}}}(\cdot, \cdot),$

$\qquad\qquad c_T = \mathcal{N}_{\frac{1}{T}\sum_{t=1}^{T}\widetilde{\mathbf{y}}_t}(\mathbf{0}, \frac{1}{T}\boldsymbol{\Delta} + \mathbf{I}_N \otimes \mathbf{K})$

Therefore, we can write eq. 18 as

$$
\begin{aligned}
p(\widetilde{\mathbf{y}}_1, ..., \widetilde{\mathbf{y}}_T | \mathbf{K}) &= \int \mathcal{N}_{\widehat{\mathbf{f}}}(\widetilde{\mathbf{y}}_1, \boldsymbol{\Delta})\mathcal{N}_{\widehat{\mathbf{f}}}(\widetilde{\mathbf{y}}_2, \boldsymbol{\Delta})...\mathcal{N}_{\widehat{\mathbf{f}}}(\widetilde{\mathbf{y}}_T, \boldsymbol{\Delta})\mathcal{N}_{\widehat{\mathbf{f}}}(\mathbf{0}, \mathbf{I}_N \otimes \mathbf{K})d\widetilde{\mathbf{f}} = \int P_T d\widetilde{\mathbf{f}} \\
&= \int \prod_{t=1}^{T} c_t \mathcal{N}_{\widehat{\mathbf{f}}}(\cdot, \cdot)\, d\widetilde{\mathbf{f}} = \prod_{t=1}^{T} c_t \int \mathcal{N}_{\widehat{\mathbf{f}}}(\cdot, \cdot)\, d\widetilde{\mathbf{f}} = \prod_{t=1}^{T} c_t
\end{aligned}
$$

Its log likelihood is

$$\log p(\widetilde{\mathbf{y}}_1, \widetilde{\mathbf{y}}_2, ..., \widetilde{\mathbf{y}}_T) = \sum_{t=1}^{T} \log c_t \tag{20}$$

$$= \sum_{t=1}^{T-1} \left[ -\frac{1}{2} \log |\frac{t+1}{t} \mathbf{\Delta}| - \frac{1}{2}(\widetilde{\mathbf{y}}_{t+1} - \frac{1}{t} \sum_{j=1}^{t} \widetilde{\mathbf{y}}_j)^\top (\frac{t+1}{t} \mathbf{\Delta})^{-1}(\widetilde{\mathbf{y}}_{t+1} - \frac{1}{t} \sum_{j=1}^{t} \widetilde{\mathbf{y}}_j) \right]$$
$$-\frac{1}{2} \log |\frac{1}{T} \mathbf{\Delta} + \mathbf{I} \otimes \mathbf{K}| - \frac{1}{2T} \sum_{t=1}^{T} \widetilde{\mathbf{y}}_t^\top (\frac{1}{T} \mathbf{\Delta} + \mathbf{I} \otimes \mathbf{K})^{-1} \frac{1}{T} \sum_{t=1}^{T} \widetilde{\mathbf{y}}_t \tag{21}$$

$$= \sum_{t=1}^{T-1} \left[ -\frac{DN}{2} \log(\frac{t+1}{t}) - \frac{1}{2} \log |\mathbf{\Delta}| - \frac{1}{2}(\sqrt{\frac{t}{t+1}} \widetilde{\mathbf{y}}_{t+1} - \frac{1}{\sqrt{t(t+1)}} \sum_{j=1}^{t} \widetilde{\mathbf{y}}_j)^\top \mathbf{\Delta}^{-1}(\sqrt{\frac{t}{t+1}} \widetilde{\mathbf{y}}_{t+1} - \frac{1}{\sqrt{t(t+1)}} \sum_{j=1}^{t} \widetilde{\mathbf{y}}_j) \right]$$
$$-\frac{1}{2} \log |\frac{1}{T} \mathbf{\Delta} + \mathbf{I} \otimes \mathbf{K}| - \frac{1}{2T} \sum_{t=1}^{T} \widetilde{\mathbf{y}}_t^\top (\frac{1}{T} \mathbf{\Delta} + \mathbf{I} \otimes \mathbf{K})^{-1} \frac{1}{T} \sum_{t=1}^{T} \widetilde{\mathbf{y}}_t \tag{22}$$

$$= -\frac{DN}{2} \log(T) + \sum_{t=1}^{T-1} \left[ -\frac{1}{2} \log |\mathbf{\Delta}| - \frac{1}{2}(\sqrt{\frac{t}{t+1}} \widetilde{\mathbf{y}}_{t+1} - \frac{1}{\sqrt{t(t+1)}} \sum_{j=1}^{t} \widetilde{\mathbf{y}}_j)^\top \mathbf{\Delta}^{-1}(\sqrt{\frac{t}{t+1}} \widetilde{\mathbf{y}}_{t+1} - \frac{1}{\sqrt{t(t+1)}} \sum_{j=1}^{t} \widetilde{\mathbf{y}}_j) \right]$$
$$-\frac{1}{2} \log |\frac{1}{T} \mathbf{\Delta} + \mathbf{I} \otimes \mathbf{K}| - \frac{1}{2T} \sum_{t=1}^{T} \widetilde{\mathbf{y}}_t^\top (\frac{1}{T} \mathbf{\Delta} + \mathbf{I} \otimes \mathbf{K})^{-1} \frac{1}{T} \sum_{t=1}^{T} \widetilde{\mathbf{y}}_t \tag{23}$$

$$= \sum_{t=1}^{T-1} \log \mathcal{N} \left( \sqrt{\frac{t}{t+1}} \widetilde{\mathbf{y}}_{t+1} | \frac{1}{\sqrt{t(t+1)}} \sum_{j=1}^{t} \widetilde{\mathbf{y}}_j, \mathbf{\Delta} \right) - \frac{1}{2\sqrt{T}} \sum_{t=1}^{T} \widetilde{\mathbf{y}}_t^\top (\mathbf{\Delta} + T\mathbf{I} \otimes \mathbf{K})^{-1} \frac{1}{\sqrt{T}} \sum_{t=1}^{T} \widetilde{\mathbf{y}}_t - \frac{1}{2} \log |\mathbf{\Delta} + T\mathbf{I} \otimes \mathbf{K}|$$

$$= \sum_{t=1}^{T-1} \log \mathcal{N} \left( \frac{1}{\sqrt{t(t+1)}} \sum_{j=1}^{t} \widetilde{\mathbf{y}}_j | \sqrt{\frac{t}{t+1}} \widetilde{\mathbf{y}}_{t+1}, \mathbf{\Delta} \right) + \log \mathcal{N} \left( \frac{1}{\sqrt{T}} \sum_{j=1}^{T} \widetilde{\mathbf{y}}_j | \mathbf{0}, \mathbf{\Delta} + T\mathbf{I} \otimes \mathbf{K} \right)$$

Thus the marginal distribution is

$$p(\widetilde{\mathbf{y}}_1, ..., \widetilde{\mathbf{y}}_T | \mathbf{K}) = \mathcal{N} \left( \frac{1}{\sqrt{T}} \sum_{j=1}^{T} \widetilde{\mathbf{y}}_j | \mathbf{0}, \mathbf{\Delta} + T\mathbf{I} \otimes \mathbf{K} \right) \prod_{t=1}^{T-1} \mathcal{N} \left( \frac{1}{\sqrt{t(t+1)}} \sum_{j=1}^{t} \widetilde{\mathbf{y}}_j - \sqrt{\frac{t}{t+1}} \widetilde{\mathbf{y}}_{t+1} | \mathbf{0}, \mathbf{\Delta} \right)$$

## Appendix B: black box variational inference

The log marginal likelihood for eq. 7 can be lower bounded by introducing any distribution over latent variable which has support where true posterior $p(\mathbf{X}|\mathbf{Y}, \mathbf{\Delta}, \theta)$ does, and then appealing to Jensen's inequality (due to the concavity of the logarithm function):

$$\log p(\mathbf{Y}|\mathbf{\Delta}, \theta) = \log \int p(\mathbf{Y}|\mathbf{X}, \mathbf{\Delta}, \theta)p(\mathbf{X})d\mathbf{X} \geq \int q(\mathbf{X}|\lambda) \log \frac{p(\mathbf{Y}|\mathbf{X}, \mathbf{\Delta}, \theta)p(\mathbf{X})}{q(\mathbf{X}|\lambda)} d\mathbf{X} \tag{24}$$

where $q(\mathbf{X}|\lambda)$ is the variational approximating distribution for the true posterior controlled by some free variational parameters $\lambda$. We assume $q(\mathbf{X}|\lambda)$ to be a standard normal distribution. In E-step, we optimize the Evidence Lower BOund (ELBO),

$$\mathcal{L}(\lambda) \triangleq \mathbb{E}_{q(\mathbf{X}|\lambda)} \left[ \log p(\mathbf{Y}|\mathbf{X}, \mathbf{\Delta}, \theta) + \log p(\mathbf{X}) - \log q(\mathbf{X}|\lambda) \right] \tag{25}$$

A standard gradient descent method can be used to maximize the ELBO over the variational parameter with analytic computation of the expectation. However, the expectation of the first term in eq. 25 doesn't have a closed-form solution. Therefore, we will employ the Black Box Variational Inference (BBVI) [20] to maximize the ELBO with stochastic optimization. The BBVI algorithm requires the computation of noisy unbiased gradients of the ELBO with Monte Carlo samples from the variational

distribution,

$$\nabla_\lambda \mathcal{L}(\lambda) \approx \frac{1}{l} \sum_{i=1}^{l} \nabla_\lambda \log q(\mathbf{X}_l|\lambda)(\log p(\mathbf{Y}|\mathbf{X}_l, \mathbf{\Delta}, \theta) + \log p(\mathbf{X}_l) - \log q(\mathbf{X}_l|\lambda)), \text{ where } \mathbf{X}_l \sim q(\mathbf{X}|\lambda). \quad (26)$$

This gradient involves calculating the log likelihood of $p(\mathbf{Y}|\mathbf{X}, \mathbf{\Delta}, \theta)$ with $(DN) \times (DN)$ covariance matrices, which is the computational bottleneck of the evaluation. However, we can efficiently evaluate it with the nice property of Kronecker product.

## Appendix C: inverting the covariance matrix

The key problem is to invert the covariance matrix $\mathbf{C} = T\mathbf{I}_N \otimes \mathbf{K} + \mathbf{\Sigma}_N \otimes \mathbf{\Sigma}_D$.

Let $\mathbf{\Sigma}_D = \mathbf{U}_D \mathbf{\Lambda}_D \mathbf{U}_D^\top$ and $\mathbf{\Sigma}_N = \mathbf{U}_N \mathbf{\Lambda}_N \mathbf{U}_N^\top$ be the eigen-decompositions of $\mathbf{\Sigma}_D$ and $\mathbf{\Sigma}_N$. The covariance matrix $\mathbf{C}$ can be factorized as

$$\begin{aligned}
\mathbf{C} &= T\mathbf{I}_N \otimes \mathbf{K} + \mathbf{\Sigma}_N \otimes \mathbf{\Sigma}_D \\
&= T\mathbf{I}_N \otimes \mathbf{K} + (\mathbf{U}_N \mathbf{\Lambda}_N \mathbf{U}_N^\top) \otimes (\mathbf{U}_D \mathbf{\Lambda}_D \mathbf{U}_D^\top) \\
&= T\mathbf{I}_N \otimes \mathbf{K} + (\mathbf{U}_N \mathbf{\Lambda}_N^{\frac{1}{2}} \mathbf{\Lambda}_N^{\frac{1}{2}} \mathbf{U}_N^\top) \otimes (\mathbf{U}_D \mathbf{\Lambda}_D^{\frac{1}{2}} \mathbf{\Lambda}_D^{\frac{1}{2}} \mathbf{U}_D^\top) \\
&= T\mathbf{I}_N \otimes \mathbf{K} + \left(\mathbf{U}_N \mathbf{\Lambda}_N^{\frac{1}{2}} \otimes \mathbf{U}_D \mathbf{\Lambda}_D^{\frac{1}{2}}\right) \left(\mathbf{\Lambda}_N^{\frac{1}{2}} \mathbf{U}_N^\top \otimes \mathbf{\Lambda}_D^{\frac{1}{2}} \mathbf{U}_D^\top\right) \\
&= \left(\mathbf{U}_N \mathbf{\Lambda}_N^{\frac{1}{2}} \otimes \mathbf{U}_D \mathbf{\Lambda}_D^{\frac{1}{2}}\right) \left(\left(\mathbf{U}_N \mathbf{\Lambda}_N^{\frac{1}{2}} \otimes \mathbf{U}_D \mathbf{\Lambda}_D^{\frac{1}{2}}\right)^{-1} (T\mathbf{I}_N \otimes \mathbf{K}) \left(\mathbf{\Lambda}_N^{\frac{1}{2}} \mathbf{U}_N^\top \otimes \mathbf{\Lambda}_D^{\frac{1}{2}} \mathbf{U}_D^\top\right)^{-1} + \mathbf{I}_N \otimes \mathbf{I}_D\right) \\
&\quad \left(\mathbf{\Lambda}_N^{\frac{1}{2}} \mathbf{U}_N^\top \otimes \mathbf{\Lambda}_D^{\frac{1}{2}} \mathbf{U}_D^\top\right) \\
&= \left(\mathbf{U}_N \mathbf{\Lambda}_N^{\frac{1}{2}} \otimes \mathbf{U}_D \mathbf{\Lambda}_D^{\frac{1}{2}}\right) \left((T\mathbf{\Lambda}_N^{-1}) \otimes (\mathbf{\Lambda}_D^{-\frac{1}{2}} \mathbf{U}_D^\top \mathbf{K} \mathbf{U}_D \mathbf{\Lambda}_D^{-\frac{1}{2}}) + \mathbf{I}_N \otimes \mathbf{I}_D\right) \left(\mathbf{\Lambda}_N^{\frac{1}{2}} \mathbf{U}_N^\top \otimes \mathbf{\Lambda}_D^{\frac{1}{2}} \mathbf{U}_D^\top\right). \quad (27)
\end{aligned}$$

The complexity of inverting the first and the third terms in eq. 27 is $O(D^3 + N^3)$. The bottleneck is now inverting the second term in eq. 27. We define new notations $\widetilde{\mathbf{K}} = \mathbf{\Lambda}_D^{-\frac{1}{2}} \mathbf{U}_D^\top \mathbf{K} \mathbf{U}_D \mathbf{\Lambda}_D^{-\frac{1}{2}}$ and $\widetilde{\mathbf{C}} = T\mathbf{\Lambda}_N^{-1} \otimes \widetilde{\mathbf{K}} + \mathbf{I}_N \otimes \mathbf{I}_D$.

The problem is reduced to inverting the matrix $\widetilde{\mathbf{C}}$. Therefore the second step is to exploit the compatibility of a Kronecker product plus a constant diagonal term with eigenvalue decomposition. Let $T\mathbf{\Lambda}_N^{-1} = \mathbf{U}_T \mathbf{\Lambda}_T \mathbf{U}_T^\top$ and $\widetilde{\mathbf{K}} = \mathbf{U}_K \mathbf{\Lambda}_K \mathbf{U}_K^\top$ be the eigen-decompositions of $T\mathbf{\Lambda}_N^{-1}$ and $\widetilde{\mathbf{K}}$. Thus,

$$\widetilde{\mathbf{C}} = T\mathbf{\Lambda}_N^{-1} \otimes \widetilde{\mathbf{K}} + \mathbf{I}_N \otimes \mathbf{I}_D = (\mathbf{U}_T \otimes \mathbf{U}_K)(\mathbf{\Lambda}_T \otimes \mathbf{\Lambda}_K + \mathbf{I}_N \otimes \mathbf{I}_D)\left(\mathbf{U}_T^\top \otimes \mathbf{U}_K^\top\right), \quad (28)$$

Finally, combining eq. 27 and eq. 28 together to get

$$\mathbf{C} = \left(\mathbf{U}_N \mathbf{\Lambda}_N^{\frac{1}{2}} \otimes \mathbf{U}_D \mathbf{\Lambda}_D^{\frac{1}{2}}\right)(\mathbf{U}_T \otimes \mathbf{U}_K)(\mathbf{\Lambda}_T \otimes \mathbf{\Lambda}_K + \mathbf{I}_N \otimes \mathbf{I}_D)\left(\mathbf{U}_T^\top \otimes \mathbf{U}_K^\top\right)\left(\mathbf{\Lambda}_N^{\frac{1}{2}} \mathbf{U}_N^\top \otimes \mathbf{\Lambda}_D^{\frac{1}{2}} \mathbf{U}_D^\top\right). \quad (29)$$

## Appendix D: more 2D latent representations for 22 odors

Figure 5: We analyzed the same dataset with t-SNE and MDS, and present the results obtained in the figures. Note that neither method is able to identify the class structure of the functional or local odor set (compared to Fig. 3 in the main paper).