[Reviews · NeurIPS 2018]

Reviewer 1



The authors develop a dimensionality reduction method to identify a low-dimensional representation of olfactory responses in Piriform cortex. Each trial is embedded in a low-dimensional space, and for each neuron a different nonlinear mapping is learned to predict firing rate from this low-dimensional embedding. The nonlinear mapping is parameterized by a Gaussian Process with a relatively smooth prior, which aids in interpretability. The authors assume and exploit Kronecker structure in the noise covariance matrix of the learned model, and describe efficient methods for variational inference. I think this method could be very useful in other experimental systems--not just piriform cortex. I wish the authors had discussed other applications of this method in the paper! I generally like this paper. The writing style is dense and technical, but clear enough to be understood by readers with an understanding of Bayesian statistics. The quality of the work and attention to detail is exemplary. My only concern with this paper is that the comparison to PCA is not well-described, and there is not a deep discussion or comparison to other approaches. My guess is that PCA was applied to a D x NT matrix of firing rates, but this doesn't seem to be stated explicitly in the text (sorry if I missed it). It also would have been nice to describe how the data were pre-processed for PCA or compare to factor analysis for a stronger baseline. Other baselines that the authors might consider include tSNE and linear discriminant analysis (LDA) -- see Cunningham & Ghahramani (2015) for other linear dimensionality reduction methods. The paper also does not discuss differences between this model and previous work on neural data analysis. They cite 9 papers in the introduction, but do not go into detail about the differences between their work and others. The main difference that I see is that this work focuses on static, evoked neural firing rates to different odor stimuli while most previous work has considered temporal dynamics. The model described in this paper is also simpler and potentially more interpretable than some of these previous models. Overall I think this is a technically sound paper and should be accepted. But I encourage the authors to develop the discussion and comparison to other methods. === Update After Rebuttal === I bumped my score up slightly - I think this is a really detailed and carefully written paper and I look forward to seeing it published. One small comment - I think that the tSNE plot in the author rebuttal may not be well-tuned and you might try tweaking the perplexity for a more fair comparison. This isn't a big problem though -- I like the author's approach much better than tSNE anyways. I think an interesting direction for future research will be to see if the nonlinear mapping learned by the authors is somehow interpretable or consistent across animals. I look forward to seeing further developments.

Reviewer 2



The authors describe the application of a Gaussian Process Latent Variable Model to neural population recordings in the Piriform cortex, obtained from mice, using a fluorescent calcium reporter. The goal of the study is to model the relations between different odorants, which presumably are encoded by the brain to lie in some smooth and continuous latent space, although the Piriform cortex itself has never been shown to have the same obvious topography as e.g. the visual or auditory cortices. The central claim of the paper is that although odorants can be "described by thousands of features in a high-dimensional chemical feature space", the "dimension of olfactory perceptual space" is unknown, and the authors propose a model that identifies a "low-dimensional space governing odor perception" learned from the neural activity caused by presenting an odor to a mouse undergoing brain imaging. Discovering a true or even merely useful latent embedding of monomolecular odors is indeed an open problem, and there is a question of whether there even exists a continuous latent space for odorants. There is a body of prior work on this topic that unfortunately was not cited, such as Koulakov 2011, Mamlouk et al 2003, Haddad et al 2008 and Iurilli et al 2017. The model of choice here is a GPLVM augmented to account for specific covariance structure among trial-to-trial noise variability per neuron, as well as covariance between neural population responses to odorant stimuli. This is modeled efficiently using a Kronecker-factored full covariance matrix. Although interesting, it is my understanding that the specific model structure and optimization procedure are not methodological innovations. Regardless, if the goal of the work is to discover useful representations of odors, comparison to alternate model formulations for building the embeddings is crucial. For instance, how well does this modeling approach compare to using logistic regression on the Dragon molecular descriptors as input? Along those lines, I particularly appreciated the study of alternate treatments of the covariance structure in the model on synthetic data, but would have preferred this to be run on the real data. Further, applying this kind of test to the choice of the RBF kernel, which likely has dramatic effects on the structure of the embedding, would provide further evidence on whether the particular modeling choices made here serve the purpose of the work. For instance, if the authors hypothesized that odors are naturally organized in a smooth embedding space, the choice of the RBF kernel artificially inflates the likelihood of finding such an embedding after fitting the model. The experimental set up was described well, but some details were missing. It would be useful to know if there is a mouse driver line, or if a non-targeted viral transfection was used, how exactly 'neuropil contamination' was removed (this is often a very ad hoc computer vision processing step that can dramatically affect data quality). Overall, I found the paper clearly written, although dense in places. Unfortunately, I do not believe that this submission contains new knowledge of interest to the NIPS community, and the scientific findings are not tested thoroughly enough in the current state of the paper to be of significance. Some specific comments: - The use of the term 'topography' is a bit confusing, as it is usually used in the neuroscience context to refer to a physical correspondence to stimulus space and the layout of a cortical area. E.g., retinotopy refers to the layout of the retina mapping physically onto visual cortex. - The prior for embedding locations is set to be an uninformative prior, due to the lack of inclusion of chemical features per odorant. However, that information is available, and would be very useful to incorporate. - No axes or labels on 2B - An existing hypothesis about how odors are encoded in the piriform cortex is that it is performing random projections through the anatomy of connections from the mitral cells to pyramidal cells in the piriform. Random projections would thus be a natural baseline to compare against. - Typo: "captures each neuron?s response" ---------------------------------------------------------------- After author's response and reviewer discussion: I was surprised to see how out-of-range this review was, and after discussion with other reviewers, have revised my score upwards. My main critical thoughts on the paper were: - If this is a methods paper, it relies heavily on a single application, doesn't present clear technical superiority over competitive alternative methods, and doesn't present sufficient technical novelty, with a small exception. The GPLVM model proposed is technically interesting and challenging, but the main features used are not new, which the authors acknowledge. The exception here, as the authors point out, is modeling multi-trial covariance structure, which is indeed a tough problem to tackle in neuroscience data. I weighted the work down because of limited technical novelty, mostly ignoring the fact that this modeling approach has not been applied widely for modeling neural spiking data. Fusing an old technique with a new application is indeed an interesting endeavor, and thoroughly useful -- if we generate new knowledge in the application space. - If this is an applications paper, we did not learn anything new about the application. The stated goal of the work is to find a smooth olfactory embedding, which practitioners in the field do hope exist and have been searching for for quite some time. Properties of a putative embedding have huge potential impact in olfactory neuroscience, and in understanding how sensory circuits in general process information from the outside world. However, the authors unfairly stack the deck in their favor -- they build a GPLVM model with a smooth priors, and do not provide the reader a way to evaluate if their embedding is superior along any other axis besides smoothness. I weighted the work down because their results in this application were presented without sufficient context and comparison, and because of the circular nature of how they defined success -- smoothness is good, so we fit a model that can likely only produce smooth embeddings. However, it was brought to my attention by fellow reviewers that non-novel but cutting-edge methods (which this paper uses) cleanly applied to interesting domains (which this paper does), even if no new knowledge is generated, is of interest to the NIPS community.

Reviewer 3



Unlike vision where we have a very good understanding of the structure of the input space, olfaction is very tricky - not only are there a lot of different receptors making the input space very high dimensional but also we completely lack intuition about the structure of the space. The manuscript proposes a sophisticated variational estimation procedure for finding a low dimensional embedding of olfactory stimuli that could help build intuitions about this space. I like this paper a lot - the idea of defining tuning functions using the embedding is clever, the technical part is solid and generally well explained, and the potential neuroscience implications are significant. A few minor comments: 1) since the model is pretty complicated, it may be useful to have the corresponding graphical model in fig1. 2) on pg4: the Kronecker tricks needed to derive the covariance are nice but very hard to parse as written. Worth thinking a bit if one can streamline the explanation for this bit. 3) It may be because of the lack of space but the explanation of the model selection bit (noise cov rank + latent dimensionality) is hard to follow. 4) If the inferred number of latents is 10 cf text or so and the first 2 dimensions already capture what we knew about the odour topology (cf fig2) what do the rest of the dimensions do? Post authors' reply: "can one get additional insight into the structure of odor space by looking at more than the first 2 dimensions?" was not really addressed in the reply. What I was really asking was about the biological interpretability of the remaining 8 latent dimensions.